# Auxology of small samples: A method to describe child growth when restrictions prevent surveys

**Maciej Henneberg**[1,2]*, **Elżbieta Żądzińska**[1,3]

**1** Biological Anthropology and Comparative Anatomy Research Unit, School of Biomedicine, The University of Adelaide, Adelaide, Australia, **2** Institute of Evolutionary Medicine, University of Zurich, Zurich, Switzerland, **3** Faculty of Biology and Environmental Protection, Department of Anthropology, University of Łódź, Łódź, Poland

* maciej.henneberg@adelaide.edu.au

**Data Availability Statement:** All relevant data are within the paper and its Supporting Information files.

**Funding:** The authors received no specific funding for this work.

## Abstract

### Background

Child growth in populations is commonly characterised by cross-sectional surveys. These require data collection from large samples of individuals across age ranges spanning 1–20 years. Such surveys are expensive and impossible in restrictive situations, such as, e.g. the COVID pandemic or limited size of isolated communities. A method allowing description of child growth based on small samples is needed.

### Methods

Small samples of data (N~50) for boys and girls 6–20 years old from different socio-economic situations in Africa and Europe were randomly extracted from surveys of thousands of children. Data included arm circumference, hip width, grip strength, height and weight. Polynomial regressions of these measurements on age were explored.

### Findings

Polynomial curves based on small samples correlated well (r = 0.97 to 1.00) with results of surveys of thousands of children from same communities and correctly reflected sexual dimorphism and socio-economic differences.

### Conclusions

Fitting of curvilinear regressions to small data samples allows expeditious assessment of child growth in a number of characteristics when situations change rapidly, resources are limited and access to children is restricted.

**Competing interests:** The authors have declared that no competing interests exist.

## Introduction

Traditionally, studies of child/adolescent growth and development use sizeable cross-sectional samples that can be analyzed by determining parameters of morphological/functional traits distributions in separate age groups, or numerous sets of longitudinal observations. There are, however, situations where children of some specific, rare groups, are not very numerous or where studies of larger samples encounter legal and logistic problems, e.g. limited contact under COVID restrictions. In such situations, dividing small samples of observations into age classes, usually of one year duration, is not efficient from the point of view of statistical analyses because sample sizes for one-year parameter estimates become small. The process of growth is a continuous one and imposition of arbitrary age groupings is not the optimal analytical approach. Since various developmental processes determining size vary their rates with age, the growth is a curvilinear function of time. Thus, a continuous curve can be fitted to a scatter of size measurements by age. The advantage of such an approach is that the sample size for parameters of the curve equals the total number of individuals measured in a large age range, eg. 6–18 years.

Since 2020 the global pandemic of COVID-19 requires limited contact of individuals in order not to spread infection and thus measuring healthy children in large quantities for purposes of assessing how certain environmental conditions influence their growth is counter indicated. However, some information on how growth is affected by the pandemic may be useful. There are small, sometimes isolated, communities where total numbers of children are counted in scores. In those situations, approaches obtaining growth descriptions from small samples of data are useful. A method often used is the non-parametric curve fitting such as locally weighted scatterplot smoothing (LOWESS or LOESS) that can be applied when samples are numerous enough [1,2]. The main disadvantage of the LOESS method is that curves it produces can only be compared visually, but not analytically [1].

In the assessment of longitudinal growth of various morphological characters a number of methods are used [3–7]. Their consideration indicates that polynomials provide good descriptions of the growth of various organisms, including human children [6], frogs [8] and plants [9]. Third degree polynomials are considered "traditional" models in statistical considerations of complex models of individual longitudinal growth [10]. Thus, a growth model using a polynomial seems to be compatible with biological nature of developmental phenomena.

One of the most widely used models of growth applicable to the entire period of progressive postnatal ontogeny–polynomial model 1 of Preece-Baines [11] uses a defined number of parameters (at least 5), values of which must be estimated irrespective of the shape of an individual's growth curve. The model does not always fit data from small populations that are either incomplete or show no clear adolescent growth spurt. Although this model does not assume *a priori* the existence of the adolescent growth spurt [12,13], its authors could fit it sufficiently well to only 57% of boys' and 74% of British girls selected from the Harpenden Growth Study longitudinal records. Brown and Townsend [14] when applying model 1 of Preece and Baines to longitudinal data from the Yuendumu community in Australia failed to fit it to 37% of girls and 29% boys. In general, parametric models are incapable of analyzing growth of all individuals since they force certain theoretical assumptions into empirical analyses. There are also no models in the literature relating to growth of morphological or functional characteristics other than height and weight.

We propose here a non-parametric method that allows to describe growth of a variety of measurable characteristics using continuous growth curves fitted to small samples of data. We test it here using samples originating from South African children and compare with some results obtained in the same way for Polish children.

## Materials and methods

The cross-sectional data were derived from the dataset collected in 1986–1995 in South Africa in a community traditionally called "Cape Coloured" though now preferring other names such as "mixed". Details of data collection, together with ethics, are described in Henneberg and Louw [15] while their specific uses in [16–19]. All data from this community were collected following approval by the Human Research Ethics Committees of the University of Cape Town and the University of the Witwatersrand in Johannesburg and the Regional Office of the Department of Health Services and Welfare of the House of Representatives of the Government of the Republic of South Africa. Through the collaboration of the principals of a number of schools administered by this Regional Office, written consent was obtained from parents (guardians) of each individual child studied. Schools distributed information sheets and consent forms in the local language (Afrikaans) to parents of each child in their care. At a day of data collection each child to be assessed presented signed by its parents (guardians) consent form and was asked to participate. Only children presenting forms signed by their parents and agreeing to this request were tested. All members of the team collecting data (academics and students) spoke the local language and clearly understood what children said.

From this dataset comprising nearly 4,000 individuals we have selected, using the random number generator, data for 50 individuals of each following group: high socio-economic status (SES) urban boys, high SES girls, low SES rural boys and low SES girls. In each case the age range was 6–20 years. Chronological (calendar) age of each individual was computed precisely as a difference in days between the date of examination and the date of birth converted to years and their decimal fractions. These data are available as S1 Dataset.

Data for each of these groups were considered as if they were a set of observations collected in a situation when only 50 children could be measured in a whole studied community. Results of fitting growth curves to these samples were then compared to cross-sectional results of the whole data set analysis [15]. For some comparisons, selected 50 observations of Polish children measured in the city of Łódź (a city of 700 000 inhabitants, located in central Poland) in 2002–2004 were used. The study was part of a research program monitoring the development of pre-school and school children [20,21]. These children were measured following the agreements with local administration of the school system (Kuratorium Oświaty), school principals, parents and children. Each child was asked for a verbal consent to the examination by a member of the anthropometric team and those who refused were not measured. All members of the team spoke local language (Polish). The entire procedure has been approved by the Committee for Bioethics of Research of the University of Łódź (KBBN-UŁ).

All anthropometric examinations conducted in South Africa were carried out by a team trained and supervised by M. Henneberg who was present throughout the time of all examinations. All anthropometric measurements collected in Poland were conducted by qualified members of the staff of the Department of Anthropology, Univeristy of Łódź according to the procedures introduced by Martin and Saller [22]. Method of taking measurements has been described in detail in Henneberg and Louw [15] p.75. Weight was measured with a portable spring scale ("Hanson") in the majority of cases and a beam balance scale less frequently. Both scales were usually present at the place of examination and the spring scale was periodically calibrated against the beam balance. Such an arrangement made weight taking faster. All participants were examined without their shoes and wearing only light clothing. A standard GPM anthropometer was used to measure the distance from the floor to the vertex to determine body height. A spreading caliper was used to measure hip width (ic-ic, biiliocristal diameter). Arm circumference was measured with an elastic tape. All measurements were taken to the nearest millimeter and recorded and processed that way in accordance with the requirements

of the SI system of measures. Grip strength was measured with a hand spring dynamometer and converted to the specific grip strength by combining it with arm circumference as described in Henneberg et al. [23]. It was expressed in Newtons per square centimeter.

## Statistical analysis

Scatterplot of data on the size of a particular anthropometric character against decimal age in each sample of 50 individuals was fitted with the third degree polynomial. This simple approach is entirely objective, assumption free, though it may be lacking sophistication concerning particularities of human growth. This lack, however, avoids the circular reasoning inherent in using prior knowledge of human growth established on samples of individuals limited with respect to geographic and socio-economic origin. It is also free from LOESS assumptions of linear or quadratic local growth in sizeable (0.25–0.50) portions of data, and of their arbitrary weighting. It allows measurement of goodness of fit by coefficient of determination that shows the fraction of the total variance explained by the polynomial. Polynomial regression equations obtained were then used to calculate their first derivatives by age, that is velocities of growth. These are continuous rather than pseudovelocities, that are differences between estimated adjacent age group average values, which in jagged growth curves may vary inconsistently.

Curve fitting was executed in Microsoft Excel 2020. Values predicted by polynomial regression equations for each year of age were compared with empirical averages for each year of age grouping in the entire cross-sectional data set using the procedure commonly applied for calculation of technical errors of measurement (TEM). In this procedure differences for each age are squared and their sum for all ages divided by twice the number of comparisons. Square root of this result indicates goodness of fit:

TEM $= (\Sigma[x_p-x_t]^2)/2N)^{0.5}$. Also, a correlation coefficient between polynomial-predicted and whole sample averages for each year was calculated.

## Results

For all studied characters of South African children–body height, weight, arm circumference, hip width and specific grip strength—polynomial curves fitted to randomly selected 50 individuals aged 6 to 18 years approximated empirical growth curves derived from the cross sectional studies of about 1000 individuals each (Figs 1 and 2).

Agreement, as measured by correlation coefficients was very close (0.95–1.00) and differences amounted to a few percentage points of each variable's size (Table 1).

Polynomial curves for boys and girls and for contrasting socio-economic conditions (SES), correctly illustrated growth differences as expected from the general knowledge of growth: curves for boys indicated greater values than those for girls and low SES curves lied below those for high SES children (Fig 2). Polish children's values were greater than those of South African children as expected from the knowledge of both heritable size differences and SES difference (Fig 2). Coefficients of determination $R^2$ of polynomial regressions can be used to assess individual variability of a given trait during growth (1- $R^2$). As can be expected, body height has less individual variability ($R^2 \sim 0.7$–0.8) than body mass or arm circumference ($R^2 \sim 0.6$–0.4), (Table 2).

First derivatives of polynomial distance curves can be considered as measures of growth velocities. Although they display generally expected increase in velocity towards puberty and then its decline, they, being parabolas, are not precise enough to characterize details of growth (Fig 2).

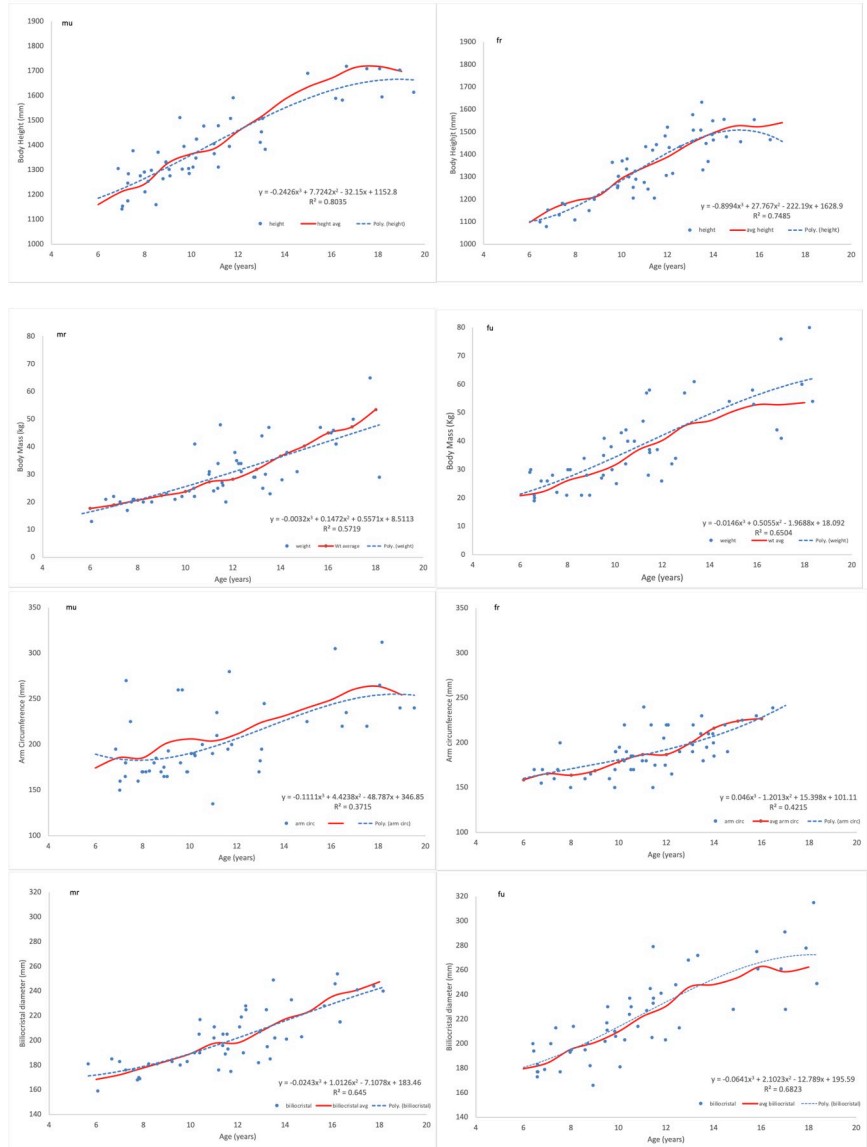

**Fig 1. Fitting of polynomial curves to characteristics of samples of 50 South African boys and girls** Upper row—body heights of urban males and rural females, second row—body masses of rural males and urban females, third row—arm circumferences of urban males and rural females, the last row—hip widths of rural males and urban females. Fitted curves are compared with curves based on one year age averages from large (N~1000) cross-sectional samples wherefrom the 50 individuals were randomly selected.

## Discussion

The curves fitted to small samples studied here show a potential to correctly characterise growth. They fit sufficiently well to cross-sectional empirical curves based on large samples and reflect expected differences between sexes, socioeconomic groups and populations. However, as unstructured models, these curves may be unstable at the extremities and their parameters do not have biological interpretations [7]. Low-order (2–3) polynomials were used to describe the early part of postnatal growth [24]. Chirwa et al. [25], comparing the fitness of four structural and two non-structural growth models using the longitudinal child growth data

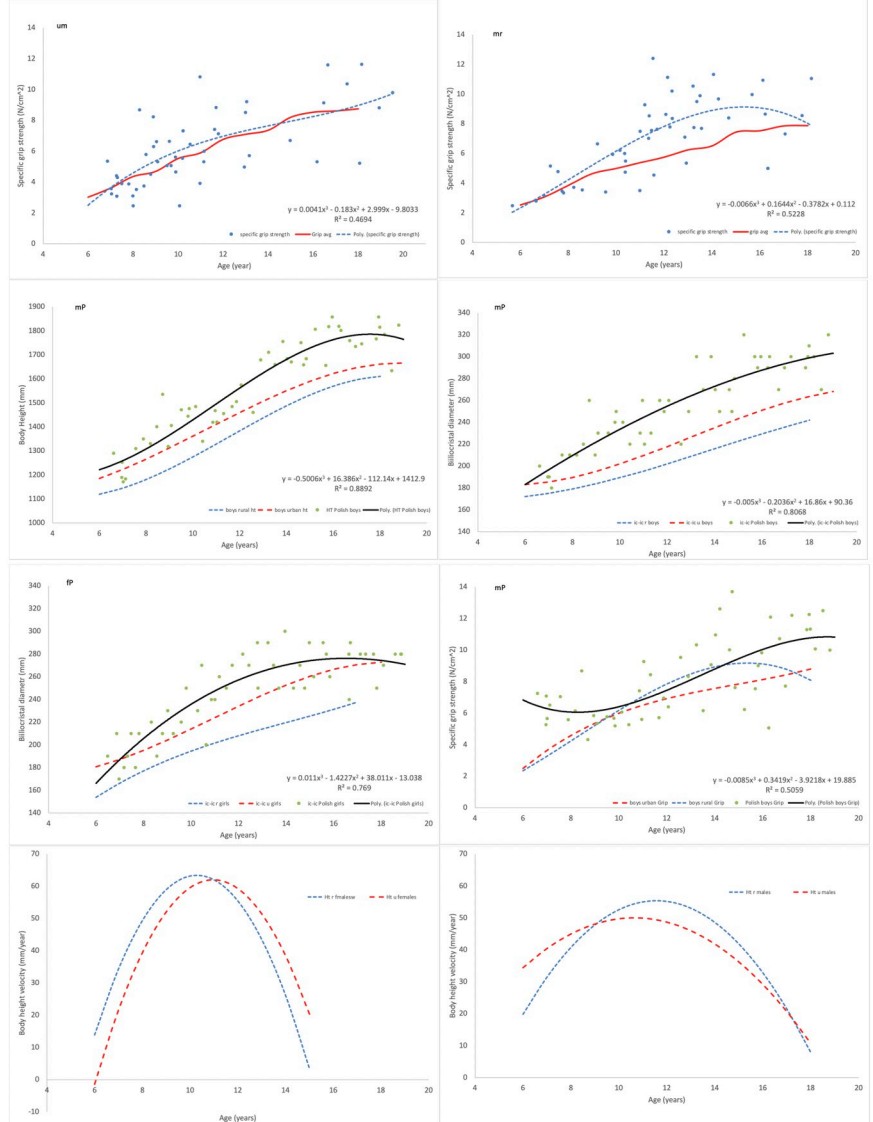

**Fig 2. Various applications of polynomial curves fitted to samples of 50 randomly selected children.** Upper row–a functional character, specific grip strength of the right hand of urban (left) and rural (right) South African males. Second and third rows–curves fitted to body height, hip width and specific grip strength of Polish children (solid lines) compared to curves for urban (long dashes) and rural (short dashes) South African children. The bottom row— velocities of height calculated as first degree derivatives of third degree polynomials fitted to 50 randomly selected South African females and males. In specific grip strength comparison note that, due to standardization on muscle cross-section (= size) there is no clear difference between urban and rural samples nor African and Polish males, as expected.

from Soweto-Johannesburg in South Africa, concluded that the 3rd order polynomial is as good as the structural Berkey-Reed 1st order model for modelling weight during infancy and childhood (up to 10 years).

Recent changes in social attitudes in developed countries together with increased considerations of research ethics make it difficult to recruit large samples of healthy children for growth studies, while local communities where child growth may be seriously compromised are difficult to approach due to conflicts or access restrictions. COVID-19 pandemic resticts contacts

**Table 1. Results of comparisons between third degree polynomial curves fitted to 50 randomly selected children characteristics and year-of-growth averages for N~1000 samples from which these 50 children were selected.**

| Characteristic | MALES | | | | FEMALES | | | |
|---|---|---|---|---|---|---|---|---|
| | rural | | urban | | rural | | urban | |
| | R | TEM | R | TEM | R | TEM | R | TEM |
| Body height (mm) | 0.99 | 15.3 | 1.00 | 24.5 | 0.99 | 21.5 | 0.99 | 23.3 |
| Body mass (Kg) | 0.99 | 1.6 | 0.98 | 2.1 | 0.99 | 1.1 | 0.99 | 2.2 |
| Arm circumf. (mm) | 0.97 | 10.0 | 0.97 | 6.4 | 0.98 | 3.5 | 0.97 | 8.4 |
| Biiliocristal (mm) | 1.00 | 2.3 | 0.99 | 3.6 | 0.98 | 5.5 | 0.99 | 7.3 |
| Grip strength(N/cm$^2$) | 0.95 | 1.0 | 0.98 | 0.3 | 0.99 | 0.9 | 0.99 | 0.9 |

R–correlation between polynomial estimates of year-of-growth averages and actual year-of-growth averages, TEM–technical error of measurement calculated as a square root of the sum of squared annual differences divided by twice the number of annual groups (6–18 years = 13 groups).

with healthy subjects. This situation may continue for a considerable time during which assessments of child growth should be done. Therefore ability to assess growth on small samples becomes useful. Growth of body height and weight of small numbers of children can be assesed by comparing individuals to WHO growth charts or similar instruments, but growth and development of other characteristics such as transverse body dimensions, circumferences or functional traits finds no comparable international standards. Fitting growth curves remains the only way to assess their growth in small samples. Assessing functional abilities of children in poor communities is more important than studying their physical size [23,26].

In the past anthropometric methods were used, incorrectly, to define "ethnic" or "racial" differences between populations. Since human anthropometric variation is continuous and reflects adaptive responses of human bodies to their immediate living conditions in addition to possible heritable differences, the results of this paper should not be used for any attempts to define taxonomic differences among human populations.

**Table 2. Coefficients of the third degree polynomial curves fitted to characteristics of randomly selected samples of 50 children aged 6–18 years compared between sexes, socioeconomic status and African and Polish samples.**

| | Males | | | | | Females | | | | |
|---|---|---|---|---|---|---|---|---|---|---|
| Characteristic | a | b | c | d | R$^2$ | a | b | c | d | R$^2$ |
| Body height, rural | -0.383 | 13.3 | -98.3 | 1314.8 | 0.77 | -0.900 | 27.8 | -222.1 | 1628.9 | 0.75 |
| Body height, urban | -0.243 | 7.7 | -32.2 | 1152.8 | 0.80 | -0.855 | 28.1 | -246.8 | 1877.6 | 0.68 |
| Body height, Polish | -0.501 | 16.4 | -112.1 | 1412.9 | 0.89 | -0.032 | -2.9 | 125.3 | 503.9 | 0.85 |
| Body mass, rural | -0.0032 | 0.15 | 0.56 | 8.5 | 0.57 | -0.015 | 0.647 | -4.42 | 23.8 | 0.59 |
| Body mass, urban | -0.0609 | 2.31 | -24.66 | 105.7 | 0.64 | -0.015 | 0.506 | -1.97 | 18.1 | 0.65 |
| Body mass, Polish | -0.0064 | 0.21 | 2.06 | 3.8 | 0.73 | 0.009 | -0.603 | 13.60 | -46.8 | 0.72 |
| Arm circ. rural | 0.072 | -2.51 | 31.8 | 39.4 | 0.37 | 0.046 | -1.201 | 15.40 | 101.1 | 0.42 |
| Arm circ. urban | -0.111 | 4.42 | -48.8 | 346.9 | 0.44 | 0.118 | -4.169 | 52.96 | -19.4 | 0.33 |
| Arm circ. Polish | 0.147 | -5.71 | 77.3 | 121.9 | 0.50 | 0.114 | -5.007 | 73.15 | -126.5 | 0.36 |
| Hip width, rural | -0.024 | 1.013 | -7.11 | 183.46 | 0.65 | 0.041 | -1.704 | 29.51 | 29.2 | 0.56 |
| Hip width, urban | -0.048 | 1.873 | -15.86 | 221.04 | 0.69 | -0.064 | 2.102 | -12.79 | 195.5 | 0.68 |
| Hip width, Polish | -0.005 | -0.204 | 16.86 | 90.36 | 0.81 | 0.011 | -1.423 | 38.01 | -13.0 | 0.77 |
| Specific grip, rural | -0.007 | 0.644 | -0.378 | 0.11 | 0.52 | 0.009 | -0.347 | 4.972 | 18.8 | 0.34 |
| Specific grip, urban | 0.004 | -0.183 | 2.999 | -9.80 | 0.47 | -0.009 | 0.284 | -2.297 | 8.9 | 0.52 |
| Specific grip Polish | -0.009 | 0.342 | -3.922 | 19.89 | 0.51 | 0.002 | -0.061 | 0.875 | 2.0 | 0.07 |

Body mass in kg, specific grip strength in N/cm$^2$, others in mm. The equation characterising each curve is: $y = ax^3 + bx^2 + cx + d$, where x- age in years.

Small sample size affects precision of results of any statistical analysis, thus the method proposed here should be used with caution, however, it is sufficient to indicate directions of differences.

## Supporting information

**S1 Dataset.**
(XLSX)

## Author Contributions

**Conceptualization:** Maciej Henneberg, Elżbieta Żądzińska.

**Data curation:** Maciej Henneberg, Elżbieta Żądzińska.

**Formal analysis:** Maciej Henneberg.

**Investigation:** Maciej Henneberg, Elżbieta Żądzińska.

**Methodology:** Maciej Henneberg.

**Validation:** Elżbieta Żądzińska.

**Visualization:** Maciej Henneberg.

**Writing – original draft:** Maciej Henneberg, Elżbieta Żądzińska.

**Writing – review & editing:** Maciej Henneberg, Elżbieta Żądzińska.

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
