## [Decision Letter · Decision Letter 0]

22 Apr 2022

PONE-D-22-04086Auxology of small samples: a method to describe child growth when restrictions prevent surveysPLOS ONE

Dear Dr. Henneberg,

Thank you for submitting your manuscript to PLOS ONE. After careful consideration, we feel that it has merit but does not fully meet PLOS ONE’s publication criteria as it currently stands. Therefore, we invite you to submit a revised version of the manuscript that addresses the points raised during the review process.

We look forward to receiving your revised manuscript.

Kind regards,

Francesco Maria Galassi

Academic Editor

PLOS ONE

Journal Requirements:

3. Our policy for research in this area aims to improve transparency in the reporting of research performed outside of researchers’ own country or community. The policy applies to researchers who have travelled to a different country to conduct research, research with Indigenous populations or their lands, and research on cultural artefacts. The questionnaire can also be requested at the journal’s discretion for any other submissions, even if these conditions are not met.  Please find more information on the policy and a link to download a blank copy of the questionnaire here: https://journals.plos.org/plosone/s/best-practices-in-research-reporting. Please upload a completed version of your questionnaire as Supporting Information when you resubmit your manuscript.

4. We noted in your submission details that a portion of your manuscript may have been presented or published elsewhere. 

(PLOS Medicine)

Reviewers' comments:

Reviewer's Responses to Questions

**Comments to the Author**

1. Is the manuscript technically sound, and do the data support the conclusions?

Reviewer #1: Yes

Reviewer #2: Yes

2. Has the statistical analysis been performed appropriately and rigorously? 

Reviewer #1: Yes

Reviewer #2: Yes

3. Have the authors made all data underlying the findings in their manuscript fully available?

Reviewer #1: Yes

Reviewer #2: No

4. Is the manuscript presented in an intelligible fashion and written in standard English?

Reviewer #1: Yes

Reviewer #2: Yes

5. Review Comments to the Author

Reviewer #1: The study proposes a unique method which is parsimonious and efficient in relation to previous methods such as parametric methods. Essentially, the non-parametric method developed describes growth of South African and Polish children using various morphometric elements.

The language of the paper is satisfactory and accessible, with the exception of a few grammatical errors. The method is clearly and logical described.

The results produced are based on the sample size are significant. Also, the study utilises continuous growth curves to a small sample size. Groups of 50 children were randomly selected from a very large cohort. The number of children is satisfactory in my opinion as it does not detract from the study’s hypothesis. The results obtained incorporate the application of polynomial curves in order to show auxological features of the groups.

The method is ideal as it considerably cuts field time and measuring large amounts of individuals which can be problematic due to limited fieldwork time, conflict/war, or bureaucratic issues. The study addresses an old but important issue of the correlation between sample size and reliable information. Does more mean better? Or is less a better approach in extrapolating tenable results? The study provides a testable hypothesis which deserves further attention in auxological studies.

Reviewer #2: This is a very good manuscript.

Only 2 points need to be addressed. They are:

1) The limitation of this study must be mentioned.

2) The utility of this paper must be highlighted.

Minor revisions are required.

6. PLOS authors have the option to publish the peer review history of their article (what does this mean?). If published, this will include your full peer review and any attached files.

Reviewer #1: No

Reviewer #2: **Yes: **Kaushik Bose

---

## [Author Response · Author response to Decision Letter 0]

11 May 2022

Response submitted as a file with responses in red font. Here we copy this file, but it will not diferentiate our responses in a colour different from the requests. Please read the submitted file rather than the text below for ease of distinguishing our replies.

Replies to the Editor and reviewers

Journal Requirements:

We have followed style requirements

We have specified types of consent for both South African and Polish children in the Methods section giving names of ethics committees and describing how individual consent, and consent of parents/guardians was obtained. See lines 90-104 and 117-122 of the revised manuscript.

3. Our policy for research in this area aims to improve transparency in the reporting of research performed outside of researchers’ own country or community. The policy applies to researchers who have travelled to a different country to conduct research, research with Indigenous populations or their lands, and research on cultural artefacts. The questionnaire can also be requested at the journal’s discretion for any other submissions, even if these conditions are not met. Please find more information on the policy and a link to download a blank copy of the questionnaire here: https://journals.plos.org/plosone/s/best-practices-in-research-reporting. Please upload a completed version of your questionnaire as Supporting Information when you resubmit your manuscript.

Both in South Africa and in Poland research was conducted in researchers’ own countries in local communities with whom researchers had personal ongoing contacts. Filled out Questionnaire is submitted.

4. We noted in your submission details that a portion of your manuscript may have been presented or published elsewhere. 

(PLOS Medicine)

NO, no portion of our manuscript was published elsewhere as far as we can remember. internet searches failed to discover any such publication. Checked PLOS Medicine – negative result. Our paper is based on data used for other studies that had aims and methods different from the current one. For this reason, materials and some characteristics of studied children have been described in a number of papers, cited in the references to the current paper. None of these papers, however, uses the method, the approach and the results produced for the current paper. No refereed conference proceeding or publication has a content similar to the current paper.

A paper with a similar title, but different contents “Auxology of small samples: new approach applied to children and adolescents in three Aboriginal communities in Australia” has been submitted to PLOS One by Dr Żądzińska in 2015 (PONE-D-15-13511) and rejected. This paper used data sets from completely different communities (Aboriginal Australians, not African and Polish) and attempted to apply a method similar to the one in the current paper, but not identical, to a different type of data (longitudinal, not cross-sectional) with the aim of characterising growth velocities, not simply the distance growth lines, as the current paper does. Introduction to this rejected paper had some sentences similar, but not identical, to those used in the current paper. Six years after the mentioned rejection, we have done a completely new work on different data sets with altered methods, and aims and results different from the rejected paper.

We now are providing a Supplementary file with all data used for the current paper, no indication that data will be available upon request is made.

a) If there are ethical or legal restrictions on sharing a de-identified data set, please explain them in detail (e.g., data contain potentially sensitive information, data are owned by a third-party organization, etc.) and who has imposed them (e.g., an ethics committee). Please also provide contact information for a data access committee, ethics committee, or other institutional body to which data requests may be sent. No need for this indication. Data are now provided in the Supplement

The anonymised data set has been uploaded as the Supplementary Information

Done, see lines 90-105 and 118-123 of the revised manuscript

Done

Refences checked

Reviewers' comments:

Reviewer's Responses to Questions

Comments to the Author

1. Is the manuscript technically sound, and do the data support the conclusions?

Reviewer #1: Yes

Reviewer #2: Yes

2. Has the statistical analysis been performed appropriately and rigorously? 

Reviewer #1: Yes

Reviewer #2: Yes

3. Have the authors made all data underlying the findings in their manuscript fully available?

Reviewer #1: Yes

Reviewer #2: No, now the file S1 dataset containing all data used is submitted

4. Is the manuscript presented in an intelligible fashion and written in standard English?

Reviewer #1: Yes

Reviewer #2: Yes

5. Review Comments to the Author

Reviewer #1: The study proposes a unique method which is parsimonious and efficient in relation to previous methods such as parametric methods. Essentially, the non-parametric method developed describes growth of South African and Polish children using various morphometric elements.

The language of the paper is satisfactory and accessible, with the exception of a few grammatical errors. The method is clearly and logical described.

The results produced are based on the sample size are significant. Also, the study utilises continuous growth curves to a small sample size. Groups of 50 children were randomly selected from a very large cohort. The number of children is satisfactory in my opinion as it does not detract from the study’s hypothesis. The results obtained incorporate the application of polynomial curves in order to show auxological features of the groups.

The method is ideal as it considerably cuts field time and measuring large amounts of individuals which can be problematic due to limited fieldwork time, conflict/war, or bureaucratic issues. The study addresses an old but important issue of the correlation between sample size and reliable information. Does more mean better? Or is less a better approach in extrapolating tenable results? The study provides a testable hypothesis which deserves further attention in auxological studies.

Reviewer #2: This is a very good manuscript.

Only 2 points need to be addressed. They are:

1) The limitation of this study must be mentioned.

Limitations were spelt out in lines 214-216 please note that transmission of the manuscript Word file seems to shift line numbers. For this reason we have highlighted by a “comment” appropriate parts of the text

2) The utility of this paper must be highlighted.

The utility has been highlighted in lines 221-226 and 228-232 please note that transmission of the manuscript Word file seems to shift line numbers. For this reason we have highlighted by a “comment” appropriate parts of the text

Since we have already mentioned the limitations and the utility of our study in the text as indicated above, we can only interpret comments of the reviewer as requiring us to put those lines under separate section subtitles. Such practice, in a short paper, would unnecessarily disrupt the flow of our brief “Discussion” that is almost entirely devoted to limitations and utility of our method.

Minor revisions are required. Done, described above

The request of the Academic Editor in consultation with the journal editors, has been satisfied by adding the following text at lines 252-256:

“In the past anthropometric methods were used, incorrectly, to define “ethnic” or “racial” differences between populations. Since human anthropometric variation is continuous and reflects adaptive responses of human bodies to their immediate living conditions in addition to possible heritable differences, the results of this paper should not be used for any attempts to define taxonomic differences among human populations.“

The authors are firmly opposed to distinguishing biological subdivisions of our species. Human variation is predominantly individual and differences among populations do not justify separating them as categories or breeds, races or subspecies. MH lived in South Africa through the end of apartheid and participated in the first free elections there.

6. PLOS authors have the option to publish the peer review history of their article (what does this mean?). If published, this will include your full peer review and any attached files.

Do you want your identity to be public for this peer review? For information about this choice, including consent withdrawal, please see our Privacy Policy.

Reviewer #1: No

Reviewer #2: Yes: Kaushik Bose

---

## [Editor Report · Decision Letter 1]

23 May 2022

Auxology of small samples: a method to describe child growth when restrictions prevent surveys

PONE-D-22-04086R1

Dear Dr. Henneberg,

We’re pleased to inform you that your manuscript has been judged scientifically suitable for publication and will be formally accepted for publication once it meets all outstanding technical requirements.

Kind regards,

Francesco Maria Galassi, MD MRSB MCSFS FRSPH 

Academic Editor

PLOS ONE
---

## [Editor Report · Acceptance letter]

30 May 2022

PONE-D-22-04086R1 

Auxology of small samples: a method to describe child growth when restrictions prevent surveys 

Dear Dr. Henneberg:

I'm pleased to inform you that your manuscript has been deemed suitable for publication in PLOS ONE. Congratulations! Your manuscript is now with our production department. 

Kind regards, 

on behalf of

Professor Francesco Maria Galassi 

Academic Editor

PLOS ONE